# Assessing the Impact of Travel Restrictions on the Spread of the 2020 Coronavirus Epidemic: An Advanced Epidemic Model Based on Human Mobility

**Xiaofei Ye** [1] , **Yi Zhu** [2] , **Tao Wang** [3,*] , **Xingchen Yan** [4] , **Jun Chen** [5] and **Pengjun Zheng** [1]

1. Ningbo Port Trade Cooperation and Development Collaborative Innovation Center, Faculty of Maritime and Transportation, Ningbo University, Fenghua Road 818#, Ningbo 315211, China; yexiaofei@nbu.edu.cn (X.Y.); zhengpengjun@nbu.edu.cn (P.Z.)
2. Faculty of Maritime and Transportation, Ningbo University, Fenghua Road 818#, Ningbo 315211, China; zhuyi19991125@163.com
3. School of Architecture and Transportation, Guilin University of Electronic Technology, Jinji Road 1#, Guilin 541004, China
4. College of Automobile and Traffic Engineering, Nanjing Forestry University, Nanjing 210037, China; xingchenyan.acad@gmail.com
5. School of Transportation, Southeast University, Nanjing 211189, China; chenjun@seu.edu.cn
* Correspondence: wangtao@seu.edu.cn

**Abstract:** Infectious disease transmission can be greatly influenced by human mobility. During the COVID-19 pandemic, the Chinese Government implemented travel restriction policies to mitigate the impact of the disease or even block the transmission chain of it. In order to quantify the impact of these policies on the number of infections and the peak time of transmission, this research modified the traditional SIR model by considering human mobility. The proposed model was validated using a Baidu Qianxi dataset and the results indicate that the number of total infections would have increased by 1.61 to 2.69 times the current value and the peak time would have moved forward by 3 to 8 days if there were no such restriction policies. Furthermore, a mixing index $\alpha$ added in the proposed model showed that the proportion of residents using public transport to travel between different areas had a positive relationship with the number of infections and the duration of the epidemic.

**Keywords:** COVID-19; coronavirus disease; dynamic of infectious disease; SIR model; travel restrictions; human mobility

## 1. Introduction

The spread of infectious diseases is influenced by various factors such as biology, environment, weather, and human activities—particularly population migration. It has been found that densely populated communities are more susceptible to epidemics like measles, smallpox, and dengue fever [1,2]. Furthermore, cities with high population mobility pose a constant threat of disease transmission, as exemplified by measles outbreaks that never seem to become extinct [3]. Given this, it is easier to manage human activities than natural factors when it comes to preventing infectious diseases. After its outbreak in Wuhan, China at the end of 2019, the coronavirus (COVID-19) rapidly spread across the globe. As of the time of writing of this paper (15 March 2020), the total number of reported infections has reached 80,931 in China and 67,760 in the rest of the world. Most of the cases in China have occurred in Wuhan and its neighboring cities in Hubei Province (Figure 1). However, the transmission was substantially reduced in China due to a series of countermeasures, one of which was travel restrictions. Beginning on 23 January 2020, Wuhan implemented a strict travel restriction policy that effectively quarantined the city by shutting down all regular intercity transport. This policy was enforced during the largest human migration period, the Spring Festival or the so-called Chunyun in Chinese

New Year, and as a result, return trips to Chunyun were reduced and intercity mobility between Wuhan and neighboring cities decreased remarkably. In the UK, in the small city of Aberdeen in Scotland, exceeding the acceptable SARS-CoV-2 infection limits defined by the government merely by fifty humans plunged the city into a full lockdown of indefinite length. The Brazilian population, vast in size, continues to suffer from incessant lockdowns. In Russia, all governmental media continue to intimidate people with the possibility that, soon, everybody will be locked down again and every citizen will pay for his/her relaxed summer rest [4]. However, these restrictions can hinder socioeconomic development, so caution is essential when implementing them.

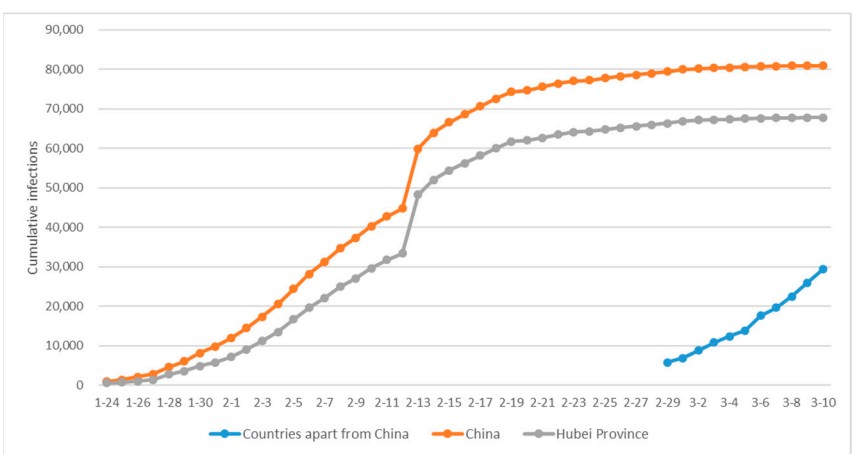

**Figure 1.** Cumulative infections of COVID-19 in China and the rest of the world; source: https://github.com/839Studio/Novel-Coronavirus-Updates (accessed on 23 March 2020).

To further understand the effectiveness of these travel restrictions, much impressive research has been conducted in this area, and some has pointed out that travel quarantines have a marked effect at the international scale, but only modestly affect the epidemic trajectory within a community [4–6]. Previous research has also revealed the importance of large-scale public health interventions [7]. Some has focused on air travel [8,9] and public travel [10], but none has quantified the effectiveness of these travel restriction policies on the total number of infections or the peak time of the pandemic. So, it is of great significance to fill this gap.

In order to reveal the effectiveness of the modelling approach on the pandemic, the classical SIR model has been modified to further address different questions in different areas [11–13]; valuable suggestions have been made but without quantified results [12]. Different assumptions may bring out different results; some research has confirmed the conclusion that the effectiveness of these lockdown measures may be overestimated because of the fact that lockdowns do not prevent any virus with droplet transmission (including SARS-CoV-2) from spreading [14]. However, most of the research in this area which is based on these classical dynamic epidemic models (such as SIR or SEIR (Susceptible–Exposed–Infectious–Recovered)) has been unable to provide quantified results to specifically clarify the effectiveness of these restriction policies. And although factors such as travel and transport are known to influence disease transmission, they are typically treated as sublevel factors that only affect the primary factors, such as transmission rates. So, it is necessary to propose a model which can help policymakers see whether it is correct to implement a mobility shutdown using specific statistics and by effectively coupling spatial mobility data with these classical dynamic epidemic models.

The aim of this study is to quantitatively analyze the effects of travel restrictions on the spread of COVID-19 by modifying the classical dynamic epidemic research model, known as the SIR (Susceptible–Infected–Removed) model, with several open-source datasets. The traditional SIR model fails to couple human mobility based on real-world data. Thus,

this paper attempts to fill these gaps by: (1) taking spatial human mobility into account to estimate the duration and number of infections; (2) proposing a novel integration methodology that links epidemic modeling with transport planning to evaluate practical policies, rather than solely focusing on improving the academic understanding of epidemic dynamics; and (3) deepening the insight into the role of public transport in the spread of the pandemic.

The remainder of this paper is organized as follows: A mobility-based SIR model is proposed in the following Section 2. Section 3 brings out the analysis based on the results of the mobility data, while the results of the model and a discussion are provided in Section 4. Finally, Section 5 concludes the paper and reveals some limitations of this research to be addressed in future research.

## 2. Mobility-Based SIR Model

The classical SIR model [14] is a compartmental model used to estimate the spread of contagious diseases. In an SIR model, individuals are classified into three groups: susceptibles, infectives, and removals. $S(t)$, $I(t)$, and $R(t)$ are used to denote the number of individuals at time $t$, respectively. It is assumed that $N(t) = S(t) + I(t) + R(t)$, where $N$ is a constant representing the total population. The model uses two parameters, $\beta$ and $\gamma$, to represent the transmission rate and the rate of removal, respectively. When susceptibles and infectives interact at time $t$, the SIR model is described by a differential equation [14]:

$$\begin{cases} \frac{dS(t)}{d(t)} = -\beta I(t) S(t) \\ \frac{dI(t)}{dt} = \frac{\beta I(t) S(t)}{N(t)} - \gamma I(t) \\ \frac{dR(t)}{dt} = \gamma R(t) \end{cases} \tag{1}$$

Whether the number of infectives increases or decreases depends on the ratio of $\beta/\gamma$, which is commonly referred to as $R0$, the basic reproduction number.

However, it misses the natural growth and death rate of people and the incubation period. To address this problem, the SEIR model was developed [7], which contains one more group, called Exposed, and takes spatial human mobility into consideration but only as a secondary factor, affecting transmission rates and delaying the spread of infection in some cases [7]. This kind of assumption does not fully capture the potential impact of human mobility from infection zones to noninfected areas, which can increase both transmission rates and the number of infectives. Reference [3] proposed a hazard model to estimate the outbreak probability of epidemics, considering intercommunity mobility. In their model, the relatively weak spatial coupling between susceptibles from community j and other communities is assumed to be binomially distributed [3]:

$$\tau \sim Bin(1, 1 - \exp(-c_j x_{t,j}, \overline{y}_t)) \tag{2}$$

where

$x_{t,j}$ denotes the proportion of susceptibles in community $j$ at time $t$;

$\overline{y}_t$ denotes the proportion of infectives in other communities at time $t$;

$c_j$ denotes the spatial coupling function between community $j$ and other communities at time $t$.

And the discrete-time hazard $h$ is the probability of the joint occurrence of an infection event and the occurrence of a spatial contact [3]:

$$h(t, j) = \frac{\beta_{u,j} S_{t,j} (1 - \exp(-c_j x_{t,j} \overline{y}_t))}{1 + \beta_{u,j} S_{t,j}} \tag{3}$$

which is an increasing function in the number of local susceptibles $S_j$, and the proportion of nonlocal individuals that are infectious, $\overline{y}_t$. It is further noted that because the

local susceptible population builds up through time, the hazard asymptotes to the spatial contact probability.

Taking human mobility data into consideration to reflect the transmission between different community, this research assumes that the value of $c_j$ is equal to $m_j^t$, which indicates the trip count from other communities to community $j$ at $t$. Interestingly, in a transport planning model, $m_j^t$ actually denotes the column sum of the inbound flow of community $j$ in an Origin–Destination (O-D) matrix. Based on the hazard model and the relationship between $S_{t+1}$, $S_t$, and $I_{t+1}$ [3,15], the mobility-based SIR model can be proposed as follows:

$$
\begin{aligned}
S_{j,t+1} &= S_{j,t} - \frac{\beta_{j,t} S_{j,t} I_{j,t}}{N_j} - \frac{S_{j,t} \sum_k m_{j,k}^t x_{k,t} \beta_{k,t}}{N_j + \sum_k m_{j,k}^t} + \sum_k m_{j,k}^t \overline{x}_t - \sum_k m_{j,k}'^t x_j \\
I_{j,t+1} &= I_{j,t} - \frac{\beta_{j,t} S_{j,t} I_{j,t}}{N_j} - \frac{S_{j,t} \sum_k m_{j,k}^t x_{k,t} \beta_{k,t}}{N_j + \sum_k m_{j,k}^t} - \gamma I_{j,t} + \sum_k m_{j,k}^t \overline{y}_t - \sum_k m_{j,k}'^t y_j \\
R_{j,t+1} &= R_{j,t} + \gamma I_{j,t} + \sum_k m_{j,k}^t \overline{z}_t - \sum_k m_{j,k}'^t z_j
\end{aligned}
\tag{4}
$$

where

$m_{j,k}^t$ denotes the trip count from $k(k \neq j)$ to $j$ (inbound flow) at time $t$;

$m_{j,k}'^t$ denotes the trip count from $j$ to $k(k \neq j)$ (outbound flow) at time $t$;

$\overline{x}_t$ denotes the proportion of $S$ in other communities $(\neq j)$ at time $t$;

$x_j$ denotes the proportion of $S$ in community $j$ at time $t$;

$\overline{y}_t$ denotes the proportion of $I$ in other communities $(\neq j)$ at time $t$;

$y_j$ denotes the proportion of $I$ in community $j$ at time $t$;

$\overline{z}_t$ denotes the proportion of $R$ in other communities $(\neq j)$ at time $t$;

$z_j$ denotes the proportion of $R$ in community $j$ at time $t$.

Note that the sum of $S_{j,t+1}$, $I_{j,t+1}$, and $R_{j,t+1}$ is not a constant. To simplify the calculation, considering that $m_{j,k}^t$ (outbound flow) and $m_{j,k}'^t$ (inbound flow) usually only have a small difference and are relatively smaller than $S$, Equation (4) can be further modified as:

$$
\begin{aligned}
S_{j,t+1} &= S_{j,t} - \frac{\beta_{j,t} S_{j,t} I_{j,t}}{N_j} - \frac{S_{j,t} \sum_k m_{j,k}^t x_{k,t} \beta_{k,t}}{N_j + \sum_k m_{j,k}^t} \\
I_{j,t+1} &= I_{j,t} - \frac{\beta_{j,t} S_{j,t} I_{j,t}}{N_j} - \frac{S_{j,t} \sum_k m_{j,k}^t x_{k,t} \beta_{k,t}}{N_j + \sum_k m_{j,k}^t} - \gamma I_{j,t} \\
R_{j,t+1} &= R_{j,t} + \gamma I_{j,t}
\end{aligned}
\tag{5}
$$

The sum of SIR $N$ is a tempo-changing variable rather than a constant, while $N(t) = S(t) + I(t) + R(t)$. This modified model assumes that intercommunity mobility influences the infection process by not only changing the transmission parameter $\beta$, but also the number of S and I directly, where the mobility variable $m_{j,k}^t$ is now an independent parameter in the model. Here, $m_{j,k}^t$ and $m_{j,k}'^t$ equal the column sum and row sum of an O-D matrix. A time-dependent O-D matrix is needed to estimate the spread of an epidemic and bridge the integration between the transport model and the epidemic dynamic model. In the next section, $m_{j,k}^t$—the outbound flow from Wuhan (in this case) to other cities in a continuous time period—is to be determined.

## 3. Human Mobility Analysis

At the time this paper was prepared, the specific location of the first reported case of COVID-19 remained unknown. However, since the first case was reported in Wuhan, it is assumed that the virus spread from Wuhan to other cities. This study focused on examining the spread of COVID-19 from Wuhan to cities within the same province (Hubei Province) and the role of human mobility (travel count) in this spread. Human mobility data, specifically the outbound flow from Wuhan to neighboring cities with and without travel restrictions, were obtained from the Baidu Qianxi study [5,16]. The former dataset was collected by Baidu, the leading IT company in China with the largest search engine

(also providing navigation services), which provides a daily intercity normalized trip index. Using a continuous duration covering the period before and after the outbreak of COVID-19, this index was calibrated with real intercity travel counts from reference [5] and historical statistical data from Chunyun released by the Wuhan Municipal Transportation Management Bureau [17] to establish a calibrated flow of human mobility. Figure 2 shows the calibrated outbound and inbound human mobility from 1 January to 7 March. Notably, Chunyun is the most significant migration event in the Chinese calendar, held from 24 January to 31 January 2020, with a high level of migration 7–15 days before the holiday as people travel to their home. On 23 January, when the travel restriction policy was put in place in Wuhan, most transportation modes from Wuhan to other cities were reduced.

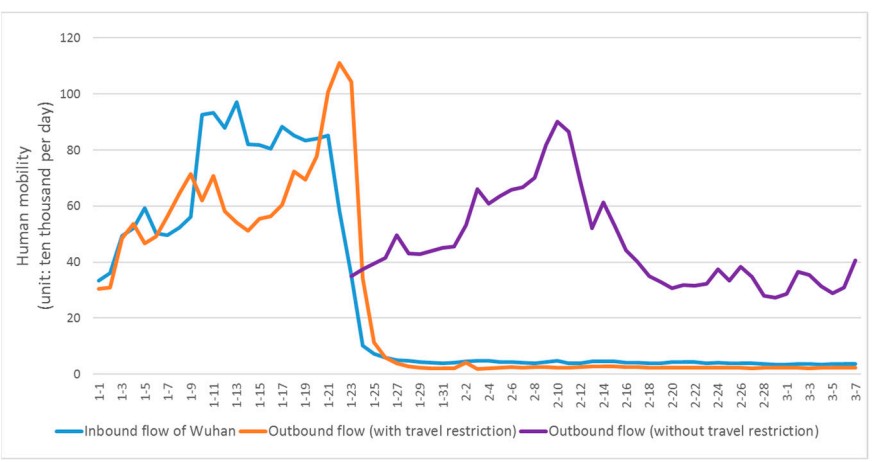

**Figure 2.** Inbound and outbound flow of people in Wuhan from January to March 2020.

The graphs in Figure 2 display the total outbound and inbound flows between Wuhan and other areas, with the orange line representing the actual outbound flow from Wuhan (with travel restrictions), the blue line indicating the actual inbound flow (with travel restrictions), and the purple line showing the hypothetical outbound flow (without travel restrictions). Prior to 23 January, Wuhan's intercity inbound and outbound travel was relatively high due to pre-Chunyun migration. As of 20 January, the inbound flow from Wuhan was greater than the outbound flow as people returned home before Chunyun. Starting from the following two days, as the COVID-19 situation became severe and rumors of a "lockdown" policy began to circulate, individuals began to rapidly leave Wuhan. This caused a sudden increase in the outbound flow, which peaked at 1.1 million individuals per day on 23 January, the day intercity transport was officially stopped. Despite the shutdown, it took five more days to reduce human mobility to a low level of less than 50,000 per day, where it remained until recently. The anticipated post-Chunyun migration, which would have mirrored the pre-Chunyun migration, did not occur due to travel restrictions. The inbound flow with these restrictions remained at around 30–40,000 individuals per day, a mere fraction of its peak value on 22 January. Interestingly, the outbound flow was even lower than the inbound flow, which aligns with reality. Although outbound travel from Wuhan was tightly controlled, exceptions were made for medical care and logistical support transport.

We formulated a hypothesis to estimate human mobility in the absence of travel restrictions, based on the distinct feature of Chunyun. Our hypothesis assumed that the inbound and outbound flows during the pre-Chunyun and post-Chunyun periods, respectively, would remain consistent with previous years. We calibrated this hypothesis using data from the previous year's Chunyun. For instance, if inbound travel primarily occurred on 22 January during Chunyun, which lasted from 24 January to 31 January, we assumed that outbound flow without travel restrictions on 1 February would be equal to the inbound flow on 22 January. Using this hypothesis, we present the hypothetical

outbound flow since Wuhan enforced travel restrictions on 23 January in Table 1 and Figure 2 (indicated by the purple line). It is apparent that the hypothetical mobility significantly differed from the actual mobility during travel restrictions, as there was no significant drop in mobility during Chunyun. After Chunyun (from 31 January), mobility remained steady at 300,000 to 400,000 per day, peaked at 900,000 per day on 10 February, and has persisted at 300,000 to 400,000 per day until today, which is approximately 15 to 20 times the actual outflow.

**Table 1.** Total outbound flow in Wuhan from 23 January to 7 March (unit: ten thousand per day).

| | Actual (with Travel Restrictions) | Hypothetical (without Travel Restrictions) |
|---|---|---|
| **Average outbound flow** | 5.66 | 46.09 |
| **Standard deviation** | 15.66 | 16.29 |
| **Max outbound flow** | 34.21 | 90.10 |
| **Date of max outbound flow** | 24 January | 10 February |

The outbound flow from Wuhan to other cities within the same province on several specific dates is shown in Table 2 and illustrated in Figures 3–6. It should be noted that these dates were selected both before and after the Chunyun holiday (24 January to 31 January). The data reveal that, during the period without travel restrictions, the outbound flow from Wuhan to neighboring cities reached its highest point between 1 February and 9 February, amounting to roughly 20 times the flow observed during the period with travel restrictions. In March, although the gap between the two scenarios narrowed, the outbound flow remained 13 times higher without travel restrictions. Notably, Xiaogan and Huanggang received the majority of outbound flow from Wuhan, which is consistent with these cities reporting the highest number of COVID-19 cases, aside from Wuhan.

**Table 2.** Outbound flow from Wuhan to neighboring cities (unit: ten thousand per day).

| Time/Scenario | Total | Max/City | Average | Standard Deviation |
|---|---|---|---|---|
| **15 January/actual** | 11.78 | 4.91/Xiaogan | 0.69 | 1.28 |
| **23 January/actual (with travel restrictions)** | 5.98 | 2.44/Xiaogan | 0.35 | 0.68 |
| **1 February/actual (with travel restrictions)** | 1.63 | 0.70/Huanggang | 0.10 | 0.17 |
| **1 February/hypothetical (without travel restrictions)** | 28.24 | 12.19/Huanggang | 1.66 | 3.07 |
| **9 February/actual (with travel restrictions)** | 1.48 | 0.63/Huanggang | 0.09 | 0.16 |
| **9 February/hypothetical (without travel restrictions)** | 17.68 | 7.65/Huanggang | 1.04 | 1.94 |
| **7 March/actual (with travel restriction)** | 0.82 | 0.32/Huanggang | 0.05 | 0.08 |
| **7 March/hypothetical (without travel restriction)** | 9.77 | 3.96/Huanggang | 0.57 | 1.00 |

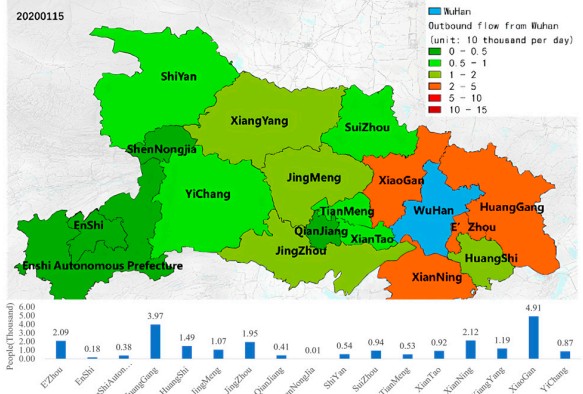 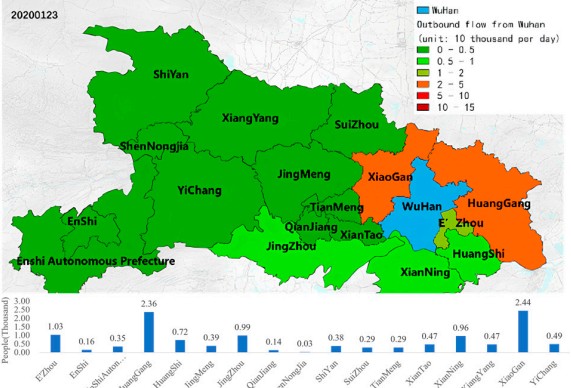

**Figure 3.** Outbound flow from Wuhan to neighboring provinces on different dates. (**Left:** 15 January; **Right:** 23 January).

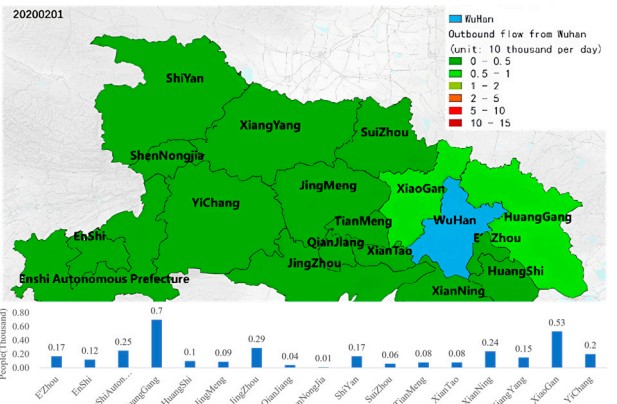 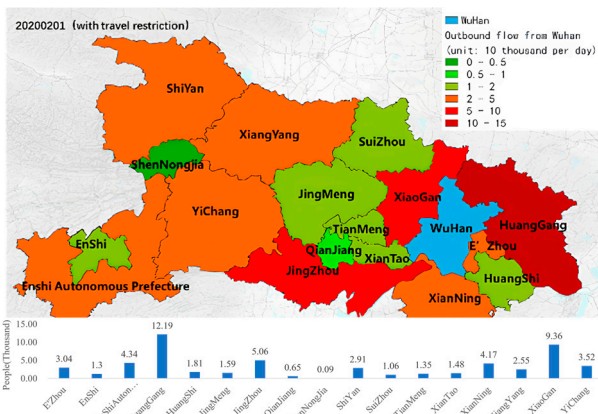

**Figure 4.** Outbound flow from Wuhan to neighboring provinces in different scenarios on 1 February. (**Left:** Actual scenario; **Right:** hypothetical scenario).

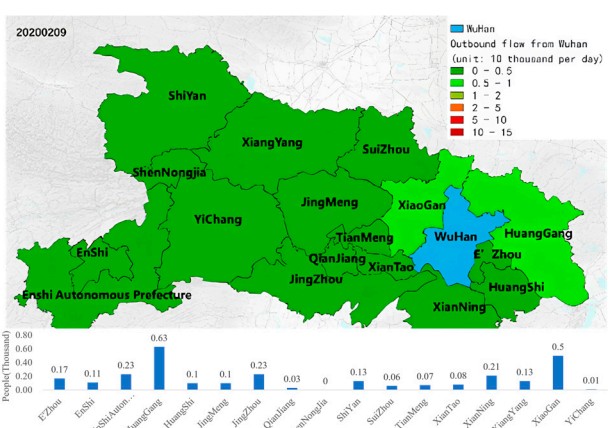 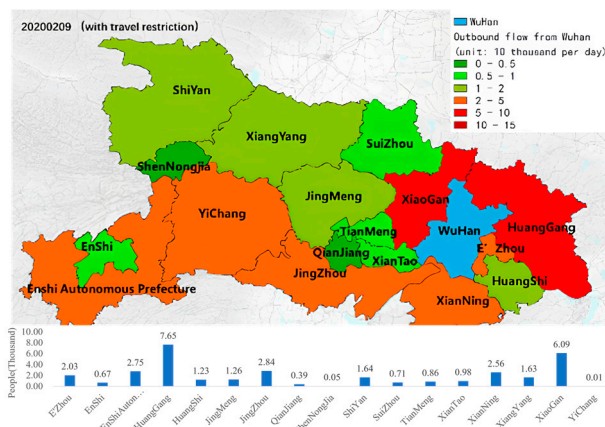

**Figure 5.** Outbound flow from Wuhan to neighboring provinces in different scenarios on 9 February. (**Left:** Actual scenario; **Right:** hypothetical scenario).

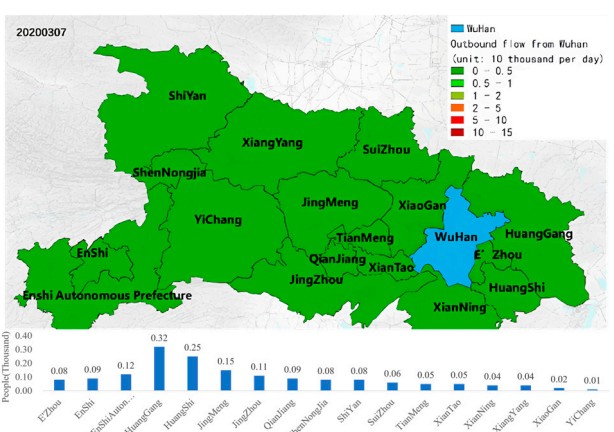 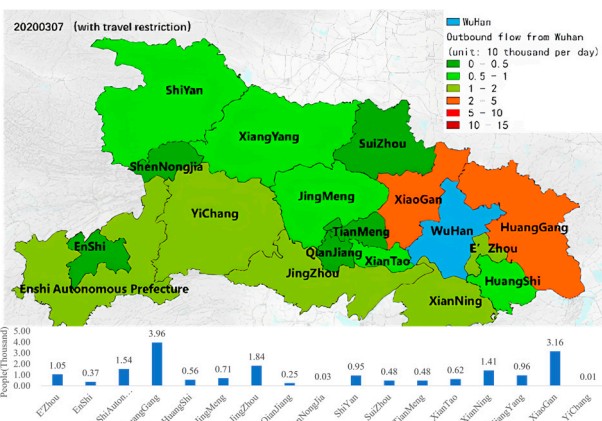

**Figure 6.** Outbound flow from Wuhan to neighboring provinces in different scenarios on 7 March. (**Left:** Actual scenario; **Right:** hypothetical scenario).

The left-hand picture in Figure 3 represents the outbound flow from the inner provinces on 15 January, at the beginning of the peak travel period before Chunyun. Travel restrictions were not in place at that time, and the travel flow was considerably high. On the contrary, as shown in the right-hand picture in Figure 3, once the restrictions were implemented, there was a significant decrease in the outbound flow from Wuhan to inner province cities, which

can be observed on 1 February (left-hand picture in Figure 4) and 9 February (left-hand picture in Figure 5). A slight rebound can be observed in the left-hand picture in Figure 6. Conversely, in the hypothetical scenario with travel restrictions, the outbound flow from Wuhan remained at a high level. The hypothetical peak flow reached 121.9 thousand in Huanggang City on 1 February, while the actual flow was only 7.1 thousand on that day.

## 4. Results and Discussion

The human mobility data mentioned above were incorporated into the mobility-based SIR model represented by Equation (5). To calibrate the model's base parameters, $\beta$, $\gamma$, and $S$ ($t = 0$), we refer to recent studies on COVID-19 [5,18–20]. As medical care conditions improved significantly and strict self-quarantine policies were enforced starting from 23 January, the transmission rate $\beta$ in our study is not a constant but a time-varying variable that follows the index distribution [20]. The estimated results of the model for two scenarios, one with travel restrictions and a hypothetical scenario without travel restrictions, are illustrated in Figures 7 and 8, respectively. The *Y*-axis depicts the cumulative number of susceptibles, infectives, and removals in all cities within Hubei Province.

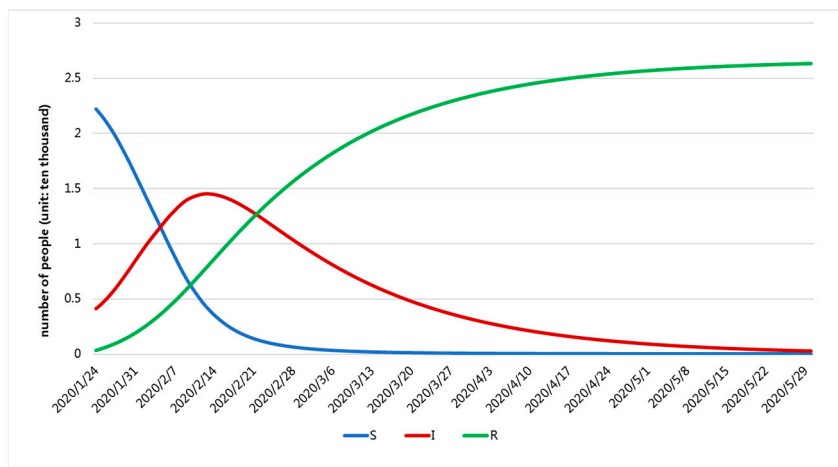

**Figure 7.** The results of the model for the scenario with travel restrictions in Hubei.

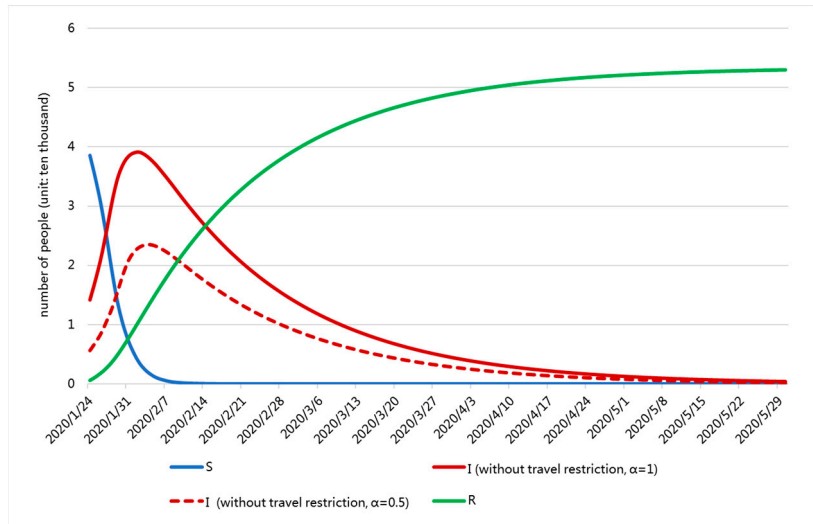

**Figure 8.** The results of the model for the scenario without travel restrictions in Hubei.

According to Figure 7, it is clear that the peak date of infectives was on 13 February, when travel restrictions were in place, which aligns with reality. Conversely, in Figure 8, the peak date of infectives occurred on 2 February in the scenario without travel restrictions.

These findings indicate that intercity travel restriction policies delayed the peak infection by 11 days; the same kind of postponement was also been confirmed in references [10,11]. This kind of short-term delay to the peak date holds significant meaning for public health emergencies as it can bring significant relief to resource shortages, giving policymakers much more time to work on relocating limited resources and, in the case of epidemics, to advance the development of vaccines and new drugs. Furthermore, by implementing restrictions on various intercity transport modes, the number of infectives would be limited to 14.5 thousand compared to 39.1 thousand if human mobility was not restricted. However, the SIR model in Equation (5) assumes that the population is evenly mixed throughout the community. Yet, the migrated population may not be mixed equally with the local population, leading to bias in Equation (5). To address this issue, a mixing index $\alpha(0 < \alpha < 1)$ was introduced to indicate the level of mixing between migratory and local populations. The index $\alpha$ can be interpreted as the proportion of nonprivate transport modes used for intercity travel or the proportion of social and relative relationships between migrants and locals. For instance, an $\alpha$ value of 1 implies that all intercity travel modes are nonprivate modes, such as trains, long-distance buses, or airplanes. To mitigate potential bias, Equation (5) can be modified as follows:

$$S_{j,t+1} = S_{j,t} - \frac{\beta_{j,t} S_{j,t} I_{j,t}}{N_j} - \frac{\alpha S_{j,t} \sum_k m_{j,k}^t x_{k,t} \beta_{k,t}}{N_j + \sum_k m_{j,k}^t}$$
$$I_{j,t+1} = I_{j,t} - \frac{\beta_{j,t} S_{j,t} I_{j,t}}{N_j} - \frac{\alpha S_{j,t} \sum_k m_{j,k}^t x_{k,t} \beta_{k,t}}{N_j + \sum_k m_{j,k}^t} - \gamma I_{j,t} \quad (6)$$
$$R_{j,t+1} = R_{j,t} + \gamma I_{j,t}$$

It is evident that Equation (6) equals Equation (5) when $\alpha$ is 1. In Figure 8, the number of infectives when $\alpha$ is 0.5 is added. The peak date for infectives ($\alpha = 0.5$) was 5 February, which was 3 days after 2 February, the peak date for the scenario with $\alpha = 1$, and 8 days earlier than the situation with travel restrictions. Furthermore, the number of infectives ($\alpha = 0.5$) was 23.4 thousand, which is 61.3% higher than the count with travel restrictions. This is reasonable and expected as the epidemic spread more rapidly in the absence of travel restrictions, resulting in more people becoming infected in a short period of time, while with travel restrictions in place, the spread of the epidemic gradually slowed down as the number of outsiders who needed to travel across regions by public transportation decreased; therefore, the peak date was delayed backwards and the total number of infected individuals decreased. Wuhan is a typical migratory city with large population migration during certain periods of time, and public transportation plays a significant role in supplying mobility demand. In line with the results of reference [11], the peak number of infections and the date of the peak time in this study increased and improved to some extent with the increase in parameter $\alpha$, which once again proved that the travel restrictions on public transportation can effectively prevent the spread of a pandemic in cities that have a similar composition of residents. Therefore, when considering policies related to mitigating the spread of an epidemic, policymakers may consider ways to suspend the sharply rising migration demand, such as providing financial support to reduce the cost of hotels in outbreak sites to provide an environment for working from home as an alternative to converting some of the traveling demand. The infective counts for the three scenarios (without travel restrictions, with travel restrictions and $\alpha = 1$, and with travel restrictions and $\alpha = 0.5$) were combined with the reported cases (Figure 9). The peak value and date for reported cases and the scenario with travel restrictions were nearly identical, but the slopes in the two scenarios are quite different. This could be due to the need for a further calibration of parameters in the SIR model by epidemic dynamic researchers. Additionally, the reported case data may not fully represent the infection itself due to undiscovered, misreported, or different statistical calibers. However, we can still observe the main trend and leave the details for further studies. According to Table 3, the duration of COVID-19 infection would have improved by late April when the number of infectives was less than

1000. However, this time frame could have been extended to early May, as shown in Table 3, if there were no travel restrictions.

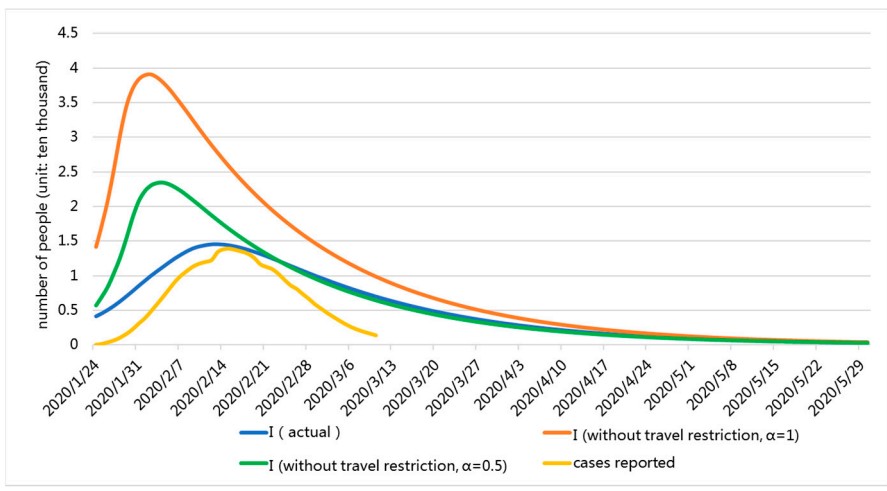

**Figure 9.** Distribution of infectives in different scenarios.

**Table 3.** Number of infectives in different scenarios.

|  | Peak Value | Relative Rate | Peak Date | Date (I < 1000) |
|---|---|---|---|---|
| **With travel restrictions** | **14,537** | **1** | **13 February** | **25 April** |
| **Without travel restrictions ($\alpha = 1$)** | 39,120 | 2.69 | 2 February | 7 May |
| **Without travel restrictions ($\alpha = 0.5$)** | 23,472 | 1.61 | 5 February | 1 May |
| **Actual cases reported** | 13,872 | 0.95 | 14 February | N/A |

## 5. Conclusions and Future Work

Drawing on data from Baidu Qianxi on human mobility and using the classic SIR model, this research proposed a mobility-based SIR model to investigate the spread of COVID-19 in Hubei Province, both with and without intercity travel restrictions. The results indicate that the strict travel restrictions significantly reduced the total and peak number of infectives, as well as delaying and reducing the overall duration of the epidemic. The main contributions of this paper are fourfold and can be summarized as follows:

(1) The traditional SIR model was modified by introducing human mobility as a primary and independent influential factor. The modified model can quantitatively analyze the impact of travel restrictions on the number of infections and peak time of the pandemic. The results from the mobility-based SIR model indicate that travel restrictions were able to delay the peak date of the COVID-19 epidemic by 11 days and also cut the number of infected individuals to nearly one third of the without-travel-restrictions scenario.

(2) Historical data from the Baidu Qianxi dataset and mobility trends resulting from Chunyun were used to estimate hypothetical travel data without restrictions. With the help of these opensource datasets, the calibrated outbound and inbound human mobility from 1 January to 7 March can be established to boost the reliability of the estimated travel data.

(3) Additionally, the methodology proposed in this research could be applied not only to intercity mobility, but also to smaller communities, districts, and larger areas such as provinces or nations, as long as human mobility exists between different regions. The value of $m_{i,j}$ in this research was equivalent to the corresponding O-D matrix unit, making this research significant in integrating both the epidemic dynamic model and the transport planning model. Governments can use the methodology proposed in this study to weigh the socioeconomic cost of implementing travel restrictions against

the benefits of reducing the number of infected individuals, delaying the outbreak, and shortening the duration of the infection.

(4) Finally, the mixing index α added in the proposed model showed that the proportion of residents using public transport to travel between different areas had a positive relationship with the number of infections and the duration of the epidemic, which can help central government leaders to pay attention to the availability of public transport when there is a public emergency. Locally, the intercommunity travel numbers can help evaluate the impact of public transport on disease transmission and aid in the decision to shut down public transport. This methodology bridges the transport model and the epidemic model, providing practical contributions for managing infectious diseases.

There are still some issues that can be addressed in future research. First, this research reveals the impact of travel restrictions on the spread of infectious diseases from the viewpoint of urban and transport planning. However, convalescent immunity, which can become much more significant as time goes by during a pandemic [21], was overlooked. Second, the study provides insights for governments considering implementing travel restrictions to slow down the spread of COVID-19, but the research area was limited to China, and the approach taken by different countries may differ based on their capacity, circumstances, and political influences. Third, while China has controlled the spread of the virus through strict travel restrictions, South Korea has not implemented city-wide lockdowns but appears to have successfully controlled the outbreak; future studies could investigate how South Korea dealt with mobility during the outbreak. Also, risk attitudes should be taken into consideration in future studies as they can affect residents' behavioral activity in response to the declaration of a pandemic, even before official government lockdowns [22]. Last, as travel restrictions can be very effective against the spread of transmission, there is ample evidence that the economy suffers and the free rights of citizens are impaired; these potential trade-offs can be detrimental to the development of a country or even a nation, which means they also deserve the attention of policymakers and can be further studied by relevant scholars.

**Author Contributions:** Conceptualization, X.Y. (Xiaofei Ye) and T.W.; methodology, X.Y. (Xiaofei Ye); software, X.Y. (Xiaofei Ye) and T.W.; validation, X.Y. (Xiaofei Ye); formal analysis, X.Y. (Xiaofei Ye), Y.Z. and T.W.; investigation, T.W.; resources, T.W.; data curation, T.W.; writing—original draft preparation, X.Y. (Xiaofei Ye) and Y.Z.; writing—review and editing, X.Y. (Xiaofei Ye) and Y.Z.; visualization, T.W.; supervision, T.W., X.Y. (Xingchen Yan), J.C. and P.Z.; funding acquisition, X.Y. (Xiaofei Ye), T.W. and P.Z. All authors have read and agreed to the published version of the manuscript.

**Funding:** This research was funded by the Transportation Technology Plan Project of Ningbo, Zhejiang (202214), Fundamental Research Funds for the Provincial Universities of Zhejiang (SJLY2023009), the National "111" Centre on Safety and Intelligent Operation of Sea Bridge (D21013), the National Natural Science Foundation of China (Nos. 71971059, 71701108, and 71861006), the National Key Research and Development Program of China–Traffic Modeling, Surveillance and Control with Connected & Automated Vehicles (2017YFE9134700), and the National Natural Science Foundation of Guangxi Province (2020GXNSFAA159153).

**Institutional Review Board Statement:** Not applicable.

**Informed Consent Statement:** Not applicable.

**Data Availability Statement:** The data used to support the findings of this study can be found through the link given below: https://qianxi.baidu.com (accessed on 23 March 2020).

**Acknowledgments:** The authors thank their mentors who gave instructions on writing this paper, and also thank the participants for their assistance in the survey.

**Conflicts of Interest:** The authors declare no conflict of interest.

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
