# Peer review of "Assessing the Impact of Travel Restrictions on the Spread of the 2020 Coronavirus Epidemic: An Advanced Epidemic Model Based on Human Mobility"

_sustainability, doi:10.3390/su151612597_

Round 1

Reviewer 1 Report (Previous Reviewer 1)

In my opinion, this paper, "Assessing the Impact of Travel Restrictions on the Spread of the 2020 Corona Virus Epidemic: An Advanced Epidemic Model based on Human Mobility," presented the right approach for solving the nonlinear differential equations epidemic model by using the probalsitic approach.

In fact, the following major comments can be addressed:

1.      According to the abstract, the manuscript is concerned with how to quantify the effect of travel limitations on the spread of COVID-19 using an enhanced epidemic model (SIR model) that takes into account human mobility using data from the Baidu Qianxi dataset. There appear to be a plethora of alternative algorithms for solving this situation. What are the benefits over those other approaches? Your model is a little behind the times.

2.      If the authors do not suggest the specified epidemic model, please provide a citation for it. Parameters and variables, along with their explanations and suggested ranges of values, should be listed in a table.

3.      Thirdly, what novel insights can be drawn from the analysis when compared to the prior art? Are these numbers accurate? There is some question about the visual portrayal.

4.      The Formula for Eq. I don't believe you since (4) you provide no evidence.

5.      The outcomes shown in the figures (8, 9 and 10) are not reproducible for me. That's why it's important to have access to the right simulation code so we can double-check the implementation.

6.      To improve the work, please provide more in-depth discussions by integrating the results of different epidemic models and numerical analyses.

7.      Format your work such that it fits the journal's guidelines, which you may get by visiting their website.

8.      There could be two data problems that need fixing. A fair IFR estimation is needed, but there is a problem with data due to unreported cases throughout the study period (important in the early stages of the pandemic when testing was less widespread). The second data problem relates to Chinese government policies, in particular given the potential underreporting of infections in 2022 (not utilised in here). However, the initial infection count may have been inaccurate due to the political climate of the pandemic in China. Considering international examples can improve dependability.

9.      Finally, I'd like to recommend giving the content a thorough examination to enhance the paper's technical quality.

The work could be rejected if the author does not respond to the above questions.

Please remove the typos/grammatical errors from he manuscript. 

Author Response

Reviewer 2 Report (Previous Reviewer 2)

The authors have made many improvements and corresponded to many of my requests. Regarding the literature, I require them to briefly discuss the issue of convalecent immunity which matters (cite, e.g. Certified Coronavirus Immunity as a Resource and Strategy to Cope with Pandemic Costs, https://onlinelibrary.wiley.com/doi/10.1111/kykl.12227). Moreover, mobility has been a constant matter in the debate on COVID. A lot of mobility reductions happened without intervention which should be acknowledged (cite, e.g. Risk attitudes and human mobility during the COVID-19 pandemic, https://www.nature.com/articles/s41598-020-76763-2, for on eof the first studies).

A final proofreading by a professional service would be good.

Author Response

Reviewer 3 Report (Previous Reviewer 3)

It is possible to observe the efforts of the authors to improve the manuscript. At various points in the manuscript the authors included recommendations suggested during the last round of review. However, the authors need to further improve the framework used. There are few references to related works in the manuscript. Therefore, before accepting the article for publication, I recommend that the authors include in the "2. Literature review" section a subsection of related works and critically present other research similar to this manuscript in the world - it does not have to be exactly the same, but on the same theme.

It is very important that the authors use references from researchers from other countries who have carried out similar studies, for example from Brazil, the United States, France, Portugal, Italy, Spain and Germany - there are similar publications that need to be discussed in this article. What did these studies do? What are your contributions? What gaps would these studies not discuss?

In the introduction, the authors could include countries that also adopted restrictive measures of human mobility - although the authors have improved the Introduction, this issue is still missing to better situate the work.

The manuscript has improved a lot, but only these small adjustments are still missing.

Needs a little revision before final version

Round 2

Reviewer 1 Report (Previous Reviewer 1)

Please check the grammatical mistakes in the revised version; otherwise, you have cooperated with all my suggestions good luck. 

Need a grammer check 

This manuscript is a resubmission of an earlier submission. The following is a list of the peer review reports and author responses from that submission.

Round 1

Reviewer 1 Report

See the attached file for the comments 

1.     I would suggest that a careful review must be taken of the text to improve the technical quality of the paper.

Reviewer 2 Report

Summary
To control the spread of COVID-19, the Chinese government implemented travel restrictions. Using an advanced SIR model that incorporates human mobility, researchers estimated the policy's effects on disease transmission. The results show a significant reduction in infections, with 1.61 to 2.69 times higher infections projected without the restrictions, and a delayed peak time of transmission.

Remarks
•    The paper should clearly highlight its novel contributions and potential surprising or counterintuitive findings. It would be useful to explicitly state the advantages of the approach over earlier studies. While it is established that travel restrictions can reduce infections, there may be trade-offs to consider. For instance, what were the costs associated with these restrictions, and did they have any long-term utility considering that most people were eventually infected or vaccinated?
•    The authors could also clarify the advantages of their method in comparison to other contributions in the literature. Since there is a significant body of literature on the topic, including studies published in non-first-tier journals, the paper should reference this literature.
•    While convalescent immunity is not relevant, of course, for the start of a pandemic, it becomes more important over time. For example, several countries implemented non-pharmaceutical interventions (NPIs) early on but then removed them progressively even without large scale vaccination, potentially due to the presence of convalescent immunity. The potential role of convalescent immunity should be considered in pandemic strategies, as quick vaccine development in the future is not guaranteed.
•    The authors should better explore whether NPIs are as crucial as they suggest to bring down infections early on or whether behavior following information campaigns is relevant. Information may be more important. In many countries, citizens reduced their mobility patterns before the implementation of NPIs (e.g. https://www.nature.com/articles/s41598-020-76763-2), suggesting that the relevance of NPIs may be overestimated. Risk attitudes matter.
•    The study should examine a longer time period. The explosion of cases in China in 2022 may also be of interest, and extending the dataset would be necessary.
•    There are potentially two data issues that need addressing. The first data issue relates to unregistered cases during the time period analyzed (that mattered early on in the pandemic when testing was less ubiquitous), which would require a reasonable IFR estimation. The second data issue concerns government policy in China, particularly as the number of reported infections in 2022 (not used in here) is likely to be incorrect. But the political nature of the pandemic in China makes it unclear whether the number of infections at the beginning of the pandemic is correct. Reliability can be increased by looking at other countries.
•    The paper could expand on the general policy implications of the analysis and highlight potential avenues for future research. This would help readers better understand the significance of the findings.

-

Reviewer 3 Report

Recommendations

[1] Introduction

Authors need to improve the introduction and clarify the objectives of the work.

[2] Theoretical Reference

Authors need to improve references. There are many works on the same theme, but the authors have explored little of this.

I recommend that authors create a related work section.

The article references are very poor. Authors need to show other works on the same theme and point out the contribution of this article.

[3] Discussions

Authors place discussions after conclusions. recommend that the discussions be moved to after the results.

The authors need to improve further the discussion. I expected a richer and more comparative discussion with what was debated in the world, particularly on issues that dealt with social distancing.

The authors need to deepen the discussion.

The quality of the text of the article is good, but it is important to review it before the final version.